Identification of non-model mammal species using the MinION DNA sequencer from Oxford Nanopore

http://orcid.org/0000-0002-9072-922X Velasquez-Restrepo Sara 1
Corrales Orozco Mariana 1
http://orcid.org/0000-0001-9144-3779 Franco-Sierra Nicolás D. 2 3
http://orcid.org/0000-0002-8906-9420 Martínez-Cerón Juan M. 1
http://orcid.org/0000-0003-3926-5244 Díaz-Nieto Juan F. 1 jdiazni@eafit.edu.co
1 Natural Systems and Sustainability Area, Universidad EAFIT , Medellín, Antioquia , Colombia
2 Syndesis Health , Palm Beach Gardens, Florida , United States
3 Corporación de Investigación e Innovación (VEDAS CII), VEDAS , Medellín, Antioquia , Colombia
Schuster Richard
Electronic publication date: 2024 Sep 25
Publication date: 2024
Volume: 12
Electronic Location ID: e17887
Received 2023 Dec 21; Accepted 2024 Jul 18
Copyright: © 2024 Velasquez-Restrepo et al.
Copyright year: 2024
Copyright holder: Velasquez-Restrepo et al.
License: This is an open access article distributed under the terms of the Creative Commons Attribution License, which permits unrestricted use, distribution, reproduction and adaptation in any medium and for any purpose provided that it is properly attributed. For attribution, the original author(s), title, publication source (PeerJ) and either DOI or URL of the article must be cited.
License URL: https://creativecommons.org/licenses/by/4.0/

Keywords: Species identification, Cryptic species, Non-model organisms, Mitochondrial genomes, Computational pipeline, Shotgun sequencing, Third generation sequencing

Funding: Universidad EAFIT Fundación Alejandro Angel Escobar, Colombia Biodiversa Cohort 2022-2 This research was funded by Universidad EAFIT through the research grant “Identificación molecular de especies en tiempo real por medio del uso de técnicas de secuenciación de tercera generación” awarded to Juan F. Díaz-Nieto. Additional funding was given to Sara Velasquez-Restrepo by Fundación Alejandro Angel Escobar, Colombia Biodiversa grant (Cohort 2022-2). The funders had no role in study design, data collection and analysis, decision to publish, or preparation of the manuscript.

==============================
Background

The Neotropics harbors the largest species richness of the planet; however, even in well-studied groups, there are potentially hundreds of species that lack a formal description, and likewise, many already described taxa are difficult to identify using morphology. Specifically in small mammals, complex morphological diagnoses have been facilitated by the use of molecular data, particularly from mitochondrial sequences, to obtain accurate species identifications. Obtaining mitochondrial markers implies the use of PCR and specific primers, which are largely absent for non-model organisms. Oxford Nanopore Technologies (ONT) is a new alternative for sequencing the entire mitochondrial genome without the need for specific primers. Only a limited number of studies have employed exclusively ONT long-reads to assemble mitochondrial genomes, and few studies have yet evaluated the usefulness of such reads in multiple non-model organisms.

Methods

We implemented fieldwork to collect small mammals, including rodents, bats, and marsupials, in five localities in the northern extreme of the Cordillera Central of Colombia. DNA samples were sequenced using the MinION device and Flongle flow cells. Shotgun-sequenced data was used to reconstruct the mitochondrial genome of all the samples. In parallel, using a customized computational pipeline, species-level identifications were obtained based on sequencing raw reads (Whole Genome Sequencing). ONT-based identifications were corroborated using traditional morphological characters and phylogenetic analyses.

Results

A total of 24 individuals from 18 species were collected, morphologically identified, and deposited in the biological collection of Universidad EAFIT. Our different computational pipelines were able to reconstruct mitochondrial genomes from exclusively ONT reads. We obtained three new mitochondrial genomes and eight new molecular mitochondrial sequences for six species. Our species identification pipeline was able to obtain accurate species identifications for up to 75% of the individuals in as little as 5 s. Finally, our phylogenetic analyses corroborated the identifications from our automated species identification pipeline and revealed important contributions to the knowledge of the diversity of Neotropical small mammals.

Discussion

This study was able to evaluate different pipelines to reconstruct mitochondrial genomes from non-model organisms, using exclusively ONT reads, benchmarking these protocols on a multi-species dataset. The proposed methodology can be applied by non-expert taxonomists and has the potential to be implemented in real-time, without the need to euthanize the organisms and under field conditions. Therefore, it stands as a relevant tool to help increase the available data for non-model organisms, and the rate at which researchers can characterize life specially in highly biodiverse places as the Neotropics.

Introduction

The Neotropics holds some of the greatest species richness and endemism in the world; however, a large fraction of such diversity is still unknown and hidden within “cryptic diversity” (Field et al., 2009; Antonelli & Sanmartín, 2011; Moreno et al., 2018; Parsons et al., 2022). For the past three decades, the recognition and description of metazoan cryptic species has been based on the complementary use of DNA sequences (with emphasis on mitochondrial DNA) and extremely detailed morphological characters (Byrne, 2023; Nachman et al., 2023). Particularly for small mammals (rodents, bats, opossums), the diagnostic characters—traits unique to a species which are vital for its identification—are almost entirely based on cranial, dental, or soft-tissue morphology that inevitably requires the collection (euthanization) of animals to be prepared as study specimens (Byrne, 2023; Nachman et al., 2023). Nevertheless, euthanizing individuals may be highly undesirable, especially in conservation scenarios such as the presence of endangered species (Kyle & Wilson, 2007; Cleman et al., 2014; Costello et al., 2016).

DNA sequencing has emerged as a vital tool to overcome the challenges associated with species identification solely based on morphological traits (Bickford et al., 2007; Schlegel et al., 2012; Lopez-Baucells et al., 2018), and in particular, mitochondrial DNA (mtDNA) has become the ideal source of data for species identification, given to its—compared to the nuclear DNA—relatively short coalescent time, small effective population size, and high mutation rate (Brown, George & Wilson, 1979; Moore, 1995). Although, within this genome, the Cytochrome Oxidase C Subunit I (COI) has gained prominence for the identification of a wide range of Metazoan organisms (Gostel & Kress, 2022; Hebert, Ratnasingham & de Waard, 2003; Hebert et al., 2003), other various mitochondrial markers (e.g., 12S, 16S, CYTB, NAD) have been historically used for species identification in particular clades (for some examples see Parson et al., 2000; Rastogi et al., 2007; Milan et al., 2020). Consequently, these multiple mtDNA markers have produced heterogeneous sequence databases (Kowalczyk et al., 2021; Antil et al., 2022), which pose a great challenge for species identification because of the absence of complete datasets of orthologues needed for comparison.

Obtaining particular amplicons for a molecular-based species identification is still commonly done using PCR-based amplification (e.g., Nowak et al., 2014; Schäffer, Zachos & Koblmüller, 2017; Pomerantz et al., 2018, 2022). However, although very useful, PCR requires the use of specific primers which can be absent for multiple non-model organisms (Schäffer, Zachos & Koblmüller, 2017; Cicero et al., 2021). In many taxonomic groups, there are primer cocktails designed for a vast range of species, but the construction of such “universal primers” relies on genomes already available in databases (usually from model organisms), where there is limited representation of Neotropical genetic diversity (Naidu et al., 2012; Schlegel et al., 2012; da Fonseca et al., 2016). Hence, in the presence of understudied species, the use of these primers can lead to the inability to obtain the DNA fragment of interest, problems of specificity, or obtaining the wrong DNA fragment (Naidu et al., 2012; Schlegel et al., 2012; da Fonseca et al., 2016).

One way to deal with the heterogeneity of mtDNA databases is to sequence the entire mitochondrial genome and have all the potential markers for the species identification regardless of the marker available in the DNA repository. Sequencing the mitogenome has been traditionally achieved through long-range PCR (Groenenberg et al., 2017), back-to-back PCRs (Emser et al., 2021) and shotgun or targeted massive parallel sequencing (Zhang, Cui & Wong, 2012; Clarke et al., 2014). Although, any of these methods has been sufficient for obtaining (routinely) mitochondrial genomes, it requires sequencing platforms (e.g., Sanger, Illumina) and molecular biology facilities which are often unavailable in developing countries (Ciocca & Delgado, 2017; Galina, Martínez & Murphy, 2023). Consequently, a typical sequencing approach involves exporting the DNA or amplicons to sequencing facilities, adding paperwork, time and budget to this process (Salager-Meyer, 2008; Rios & de Regules, 2022; Zarog, 2023). The cumulative impact of the mentioned problems hinders accurate species identification using DNA sequences, affecting the efficiency of data collection (i.e., DNA sequences) and the expansion of databases (Rios & de Regules, 2022; Zarog, 2023).

The development of the third-generation sequencing platform of Oxford Nanopore Technologies (ONT), has resulted as a promising alternative to obtain sequences on-site without the need of complex facilities. In fact, the portable MinION sequencing platform has already been used to shotgun-sequence—without the need of primers—complete mitochondrial genomes in multiple taxonomic groups such as invertebrates (Gan et al., 2019; Baeza, 2020; Filipović et al., 2022), vertebrates (Wiley & Miller, 2020; Kryukov et al., 2020; Malukiewicz et al., 2021; Baeza & García-De León, 2022), fungi (Sa et al., 2020) and plants (Jackman et al., 2020). Furthermore, ONT has emerged as a viable alternative for species identification, primarily within a DNA barcoding framework, which involves the sequencing of PCR-based amplicons, and it has already proven to be effective in diverse taxonomic groups including bacteria and viruses (Benítez-Páez, Portune & Sanz, 2016; Riaz et al., 2021), plants (Antil et al., 2022), invertebrates (Kneubehl et al., 2022) and vertebrates (Krehenwinkel, Pomerantz & Prost, 2019; Srivathsan et al., 2019, 2021; Johri et al., 2019). Notably, in recent years a few studies have shown the capabilities of this sequencing technology for real-time and onsite species identification (Parker et al., 2017; Pomerantz et al., 2018, 2022; Maestri et al., 2019).

Some studies have already used ONT shotgun sequencing reads for species identification, offering an alternative approach to circumvent the challenges associated with the amplification of specific markers (Baeza, 2020; Imai et al., 2020; Baeza & García-De León, 2022; De Vivo et al., 2022). These studies mostly provide identifications implementing phylogenetic analyses, but at least one study uses ONT reads followed by a custom-made bioinformatic pipeline to provide species-level identification of a non-model mammal species (Franco-Sierra & Díaz-Nieto, 2020). Despite the advancements that these projects made in the field of ONT-based species identification, they usually include one or few individuals of closely related species, and they have failed to test their laboratory and computational methods with multiple individuals across a broader taxonomical sampling (different families or orders). Also, these studies rely on expensive flowcells (ca. 900 USD) with massive sequencing capacity, which although generate the amount of data needed for species identification, the benefit to cost ratio for a small mitochondrial genome (ca. 17 kb) can be considered extravagant. Currently, Oxford Nanopore Technologies have produced a small-essay sequencing flowcell, the Flongle, which is available at a reduced cost (compared to the regular flow cells) and offers a remarkable level of precision with a mere 5% chemical error rate in the R9 series. Our research aims to evaluate the feasibility of using ONT shotgun reads obtained with the Flongle sequencing flowcell for the identification of multiple species of non-model organism of mammals.

Materials and Methods

Fieldwork

We conducted fieldwork in five different localities near the city of Medellín so we could rapidly process tissue samples of captured specimens. These localities include cloud mountain forest, low tropical mountain forest, tropical dry forest, and urban ecosystems (Fig. 1 and Appendix 1). We sampled each locality for 8 days implementing the trapping methodologies specific for each focal group (bats, rodents, and marsupials) proposed by Voss & Emmons (1996). For volant mammals (bats) we used eight nets of three different sizes (6 m × 3 m, 9 m × 3 m, and 12 m × 3 m) per night. For terrestrial small mammals, at each locality, we used 100 standard Sherman folding live traps (80 cm × 90 cm × 230 cm) per night. Finally, we baited traps every morning with a mixture of oats, banana, and vanilla essence, or with a single slice of banana.

Figure 1 Map of the northern section of the Cordillera Central in Colombia showing the localities of our fieldwork.

Map made by the authors, using the QGIS program (https://www.qgis.org/en/site/index.html) and spatial data from DIVA-GIS (http://diva-gis.org/). Both repositories are Free and Open Source Geographic Information System.

To validate the results obtained from our species identification pipeline (see section “Species identification pipeline with ONT raw reads”), it is essential to compare them against an accurate species identification, which in our case is based on morphology and phylogenetic systematics. Therefore, it was still necessary to collect the individuals and deposit them as voucher specimens in a museum collection. We followed all the methods of capture and handling approved by the American Society of Mammalogists Animal Care and Use Committee (Sikes & Bryan, 2016). Specimens selected for collection were sacrificed using cervical dislocation or exposure to high concentrations of CO2 (carbon dioxide displacement rate of 60% volume per minute over a duration of 2 min), in accordance with the AVMA Guidelines for the euthanasia of animals: 2020 edition (American Veterinary Medical Association (AVMA), 2020). Cervical dislocation was used primarily for bats, while terrestrial mammals were euthanized with CO2. For the morphological identification, we used specialized literature for each group as needed. All the material collected is deposited at the biological collection of Universidad EAFIT (Medellín, Colombia). All the methodological procedures of this project were approved by the Ethics Committee of the Universidad EAFIT. The approval of the field work permit was granted by the National Environmental Licensing Authority (ANLA) through Resolution No. 1160 of 2023.

Laboratory work and Oxford Nanopore Sequencing

For the DNA extraction we removed six millimeters from the distal section of the tail of rodents and opossums, and three patagium (wing membrane) samples with a 3 mm dermatological punch for bats. We washed each tissue sample with molecular biology grade ethanol to remove external contaminants. Subsequently, we performed whole genome DNA extraction using the DNeasy Blood and Tissue Kit (Qiagen Venlo, The Netherlands) following manufacturer’s instructions, and measured DNA concentration by fluorometry using the dsDNA BR assay from Qubit 3.0 (Thermo Fisher Scientific, Waltham, MA, USA). Next, for Whole Genome Sequencing, we used the MinION device with Flongle flow cells (FLO-FLG001, R9.4.1 chemistry). Library preparation was the same for all samples using the Ligation Sequencing Kit SQK-LSK110 following the Genomic DNA by Ligation SQK-LSK110-Flongle Nanopore Protocol (version GDE_9108_v110_revL_10Nov2020). Noteworthy, we conducted the adapter ligation clean-up step using LFB buffer to enhance the recovery of longer DNA fragments (>3 Kb). For the library loading step, we used 6 μl of DNA library in addition to the sequencing Buffer II (SBII) and Loading Beads II (LBII) from the Flongle Expansion kit which are stored in glass vials. We loaded the library (sequencing mix) dropwise onto the sample port ensuring that the mix entered the Flongle cell by capillary action. We controlled the MinION sequencing run from a Dell laptop (processor Intel® Xeon® CPU E3-1505M v5 @ 2.8 Hz × 4 and 32 Gb ram) using the MinKNOW software v21.05 setting a runtime of 24 h for each sample without live basecalling. After sequencing, we basecalled the raw signal (FAST5 files) using the ONT Guppy Sequencing Pipeline software 5.0.11 to obtain sequencing reads in FASTQ format. Subsequently, we used the sequencing summary file from the basecalling process to obtain sequencing performance statistics using NanoPlot 1.39 (De Coster et al., 2018). We filtered the data from each run by quality with NanoFilt 2.8.0 (De Coster et al., 2018) and only retained the reads with Q-score ≥ 7.

Mitogenome reconstruction

The reconstruction of mitochondrial genomes was conducted under two distinct scenarios (Fig. 2). First, when there is no prior knowledge about the species identity of the organism, we implemented the mitochondrial genome reconstruction based on Whole Genome Sequences. Second, when there is prior information available on the identification of the organism (e.g., order or family)—and more importantly—when there is availability of closely related reference mitogenomes in databases, we used a mitochondrial-filtered approach. For this latter approach we filtered the mitochondrial reads implementing both Minimap2 (Li, 2018) and the mtBlaster script (Franco-Sierra & Díaz-Nieto, 2020). For both scenarios, we applied two distinct assembly methods: a de novo assembler implemented in Flye 2.9 (Kolmogorov et al., 2019), and a reference-based assembler implemented in Rebaler 0.2.0 (available at https://github.com/rrwick/Rebaler). We used the --meta flag for Flye which has been designed for metagenome/uneven coverage assemblies and is able to recover different genomes within the same sample, an ideal fit for the mitochondrial/nuclear genome uneven scenario obtained with the WGS data (Kolmogorov et al., 2019, 2020). As Rebaler is a reference-based assembler, we used as reference the mitochondrial genome from the most closely related species available in the NCBI. This selection was based on the initial morphological identifications of each species. In total, we implemented six (6) different routines based on the data type (WGS and mtDNA-filtered) and assembly method (Flye and Rebaler) (Fig. 2). The first two pipelines started from WGS data, referred to as WGS (Fig. 2, pipelines 1 and 2), and a second set of pipelines used mitochondrial-filtered reads, referred to as Minimap2-filtered and mtBlaster-filtered (Fig. 2, pipelines 3–6). The name given to the resulting mitochondrial genome assemblies indicates the data type and assembler used to reconstruct them. For example, if a genome was reconstructed using mitochondrial data filtered with Minimap2 and subsequently assembled with Flye, the resulting genome would be labeled as Minimap2+Flye.

Figure 2 Computational workflow implemented for the reconstruction of mitochondrial genomes from ONT shotgun sequencing data.

Pipelines 1 and 2 use WGS reads and use de novo (Flye) and reference based (Rebaler) assemblers. Pipelines 3 to 6, initially filter the mitochondrial reads implementing Minimap2 (pipelines 3 and 4) or mtBlastr (pipeline 5 and 6) to subsequently execute the assemble step (Flye and Rebaler). The final steps (below red triangle) that include annotation and polishing are common to all six pipelines.

We annotated the resulting assemblies (final assembly generated by either Flye or Rebaler, “draft assembly”) using the command line version of MitoZ v2.4.1 (Meng et al., 2019), then we polished the annotated genomes using NanoPolish 0.14.0 (Loman, Quick & Simpson, 2015), and finally re-annotated the polished genomes (“annotated polished assemblies” in Fig. 2). We evaluated the quality of all the annotated mitogenomes (polished and not polished) using two distinct assembly criteria: hits and errors. Assembly hits correspond to complete coding genes with open reading frames (complete genes) and complete non-coding genes (complete rRNA, tRNA). On the other hand, assembly errors are coding genes with premature stop codons (denoted as stop codons), incomplete genes (incomplete), absent genes within an assembly (null), and not assembled genomes (not assembled). We aligned each mitochondrial genome assembly to a closely related reference genome (the same used for Rebaler) to evaluate the hit and error criteria (Tables S2–S5).

Species identification based on phylogenetic analyses

Because it is essential to demonstrate the accuracy of the identifications obtained with the species identification pipeline (see below), it was necessary to provide a reliable identification with an independent line of evidence, which in our case was done with phylogenetic inference analyses. For this purpose, we extracted the mitochondrial DNA markers COI and Cytb—which are widely implemented for species identification in mammals (Parson et al., 2000; Hebert, Ratnasingham & de Waard, 2003)—from each of the annotated mitochondrial genomes, but we only extracted those sequences with sufficient quality (i.e., Open Reading Frame, ORF). Then, we downloaded all the available sequences from GenBank for each of these two loci and constructed datasets for each genus, consequently, each dataset includes as ingroup all the GenBank sequences available for a specific marker within a particular genus, along with our extracted sequences (Table S1, File S2). We removed all duplicate sequences with the Dedupe algorithm implemented in Geneious 9.1.8 (Kearse et al., 2012) and used a sequence from a species of the sister genus as outgroup (Table S2). We aligned each dataset using MUSCLE (Edgar, 2004) and conducted independent gene tree analyses for each locus (COI and Cytb) based on a Maximum Likelihood approach. The best-fitting nucleotide substitution model for each alignment was determined using the Bayesian Information Criterion (BIC) and nodal support was evaluated with 1,000 bootstrap replicates using the Ultrafast Bootstrap Approximation (Hoang et al., 2018). All the phylogenetic analyses were implemented in the IQ-TREE software v2.2.0 beta COVID-edition (Nguyen et al., 2015). The resulting trees were visualized and edited using iTOL V5 (Letunic & Bork, 2021).

Species identification pipeline with ONT raw reads

For an automated species identification using the ONT raw reads, we followed the methodological approach of Franco-Sierra & Díaz-Nieto (2020) with some minor modifications. First, to obtain exclusively mtDNA reads from our WGS, we ran a blastn search using BLAST 2.13 (Camacho et al., 2009) implemented in the mtBlaster pipeline (https://github.com/nidafra92/squirrel-project/blob/master/mtblaster.py) using a mitochondrial genome database of ten model organisms of Metazoa (see Franco-Sierra & Díaz-Nieto, 2020). We defined an identity cutoff value of 90% based on an identity threshold of 95% to safeguard homology at the nucleotide level plus the average ONT error rate of ~5% (for the R9 ONT chemistry). Subsequently, we blasted the extracted mitochondrial reads against a new custom-made database containing all (6,620,011) metazoan mitochondrial sequences available in GenBank as of March 2023. For the BLAST search, the strictness of the analysis ranged between medium to high level by setting the parameters as follows: max_target_seqs =1, word_size = 11, gapopen = 1, gapextend = 2, penalty = −4, reward = 1, max_hsps = 1, perc_identity = 90, and task = blastn.

We simulated three different hypothetical scenarios, where the size of the search database (custom database of mitochondrial sequences) varied depending on the amount of taxonomic information available for each individual: (a) when no information is available (complete Metazoa mtDNA dataset), (b) when sample identity is known at the order level, and (c) when sample identity is known at the family level. For each individual we conducted three independent searches, simulating all three scenarios (Metazoa, order, and family). To enhance the time-efficiency of BLAST searches, we implemented the program DC-BLAST (Yim & Cushman, 2017). For the order and family-level search scenarios, we refined the complete Metazoa database informing the DC-BLAST program with a text file containing a comprehensive list of NCBI Taxonomy IDs, specific to each particular taxonomic level. Briefly, for flying mammals all specimens are part of the order Chiroptera and families Molossidae, Phyllostomidae, and Vespertilionidae (Table 1); non-flying small mammals are contained in the orders Didelphimorphia and Rodentia, and families Didelphidae and Cricetidae, respectively (Table 1). For each level of analysis, we also evaluated the time efficiency of DC-BLAST. We implemented all species identification analyses in the computational cluster Apolo Scientific Computing Center (Universidad EAFIT) using the DC-BLAST (Divide and Conquer BLAST for HPC) utility (Yim & Cushman, 2017). The setup used for executions consisted of two HPC nodes with 32 cores per node, 64 GB RAM per node, Intel® Xeon® CPU E5-2683 v4 @ 2.10 GHz processor. After obtaining the DC-BLAST results for all experiments, we ranked the hits for each individual (reads from each mammal collected) at each level of analysis (Metazoa, order, family) using two key parameters: first in ascending order of their e-values, followed by percentage of identity in descending order (File S1). We consider the first hit in each analysis as the most reliable identification.

Table 1 Morphological identification, locality, DNA extraction characteristics, and ONT sequence performance statistics of all specimens sequenced in this study.

EAFIT collection number	Morphological ID	Locality	Tissue	Concentration of DNA
extraction (ng/μl)	Concentration of DNA
library (ng/μl)	Number of cell pores	Sequencing run yield
(Mbp)	Mean read length (Kbp)	Number of reads	Number of reads ≥ Q–7	% Reads ≥ Q–7	% mtDNA (Minimap2)	% mtDNA (mtBlaster)	
EAFIT-M 2126	Anoura caudata	Hacienda la Sierra	P (2)	13	7.44	1	6.96	7.24	962	744	77.34	–	–	
EAFIT-M 2106	Myotis riparius	Hacienda la Sierra	P (2)	7	21.4	54	265.13	5.23	50,689	33,947	66.97	0.02	0.02	
EAFIT-M 2143	Nephelomys pectoralis	Hacienda la Sierra	T	33.8	24.2	38	251.67	3.92	64,068	56,059	87.5	0.07	0.08	
EAFIT-M 2107	Histiotus humboldtii	Hacienda la Sierra	P (2)	11.4	6.88	85	133.35	3.73	35,714	30,019	84.05	0.06	0.04	
EAFIT-M 2124	Rhipidomys latimanus	Hacienda la Sierra	T	76	21.6	66	720.86	2.96	243,742	212,128	87.03	0.05	0	
EAFIT-M 2116	Handleyomys alfaroi	Hacienda la Sierra	T	78.2	17.3	45	195.09	2.6	75,000	57,366	76.49	0.03	0.04	
EAFIT-M 2160	Sturnira bogotensis	La Miel	P (2)	13.7	7.34	55	615.41	3.04	202,206	170,523	84.33	0.02	0.04	
EAFIT-M 2157	Carollia brevicauda	La Miel	P (3)	39.8	6.66	57	277.84	2.86	88,680	69,111	77.93	0	0.03	
EAFIT-M 2164	Anoura geoffroyi	La Miel	P (3)	54.4	51	69	752.48	2.97	253,665	208,819	82.32	0.09	0.08	
EAFIT-M 2158	Artibeus lituratus	Universidad EAFIT	P (3)	102	9.86	70	271.02	2.3	118,000	101,131	85.7	0.04	0.03	
EAFIT-M 2167	Glossophaga soricina	Universidad EAFIT	P (3)	22	42	18	178.15	3.64	51,355	38,791	75.54	0	0	
EAFIT-M 2166	Artibeus lituratus	Universidad EAFIT	P (3)	43.4	63.6	–	653.56	2.66	245,545	185,909	75.71	0.02	0.02	
EAFIT-M 2165	Sturnira bogotensis	La Miel	P (3)	58.6	34.8	39	258.84	3.15	82,287	76,768	93.29	0	0.03	
EAFIT-M 2168	Carollia brevicauda	La Miel	P (3)	28.6	33	88	562.47	2.95	190,895	152,744	80.01	0.01	0.02	
EAFIT-M 2151	Nephelomys childi	Finca Cyathea	T	114	29.6	94	921.72	2.08	443,460	370,536	83.56	0.08	0.06	
EAFIT-M 2159	Myotis caucensis	Universidad EAFIT	P (3)	16.2	15.4	76	535.48	3.8	140,848	116,425	82.66	0.01	0.01	
EAFIT-M 2150	Nephelomys childi	Finca Cyathea	T	41	49.8	92	920.81	3.97	231,989	194,087	83.66	0.09	0.06	
EAFIT-M 2152	Molossus molossus	Universidad EAFIT	P (3)	41	69.6	92	701.68	3.81	184,279	152,350	82.67	0.1	0.06	
EAFIT-M 2161	Akodon affinis	Finca Cyathea	T	26.6	42.8	90	755.79	1.66	456,585	357,353	78.27	0.04	0.06	
EAFIT-M 2153	Artibeus lituratus	Finca Socolao	P (3)	37.5	7.06	0	0	0	0	–	–	–	–	
EAFIT-M 2154	Phyllostomus discolor	Finca Socolao	P (3)	46.9	8.2	0	0	0	0	–	–	–	–	
EAFIT-M 2156	Marmosa isthmica	Finca Socolao	T	70.8	27.2	26	150.51	2.53	59,398	48,630	81.87	0.38	0.28	
EAFIT-M 2155	Marmosa isthmica	Finca Socolao	T	84.2	65.4	72	523.36	3.33	157,390	123,681	78.58	0.41	0.28	
EAFIT-M 2162	Zygodontomys brevicauda	Finca Socolao	T	104	40.4	55	530.42	3.03	175,153	142,932	81.6	0.04	0.03	
Note:

Under the column Tissue samples, P makes reference to bat patagium and the number in parentheses are the dermatological punches sampled; T corresponds to tail of terrestrial mammals.

Results

Field work, laboratory work and Oxford Nanopore Sequencing

We collected a total of 24 individuals, representing 18 species, five families, and three orders (Table 1). Our DNA extractions produced an average concentration of 35.7 ng/μl (7–102 ng/μl; 17 samples) for bat patagium tissue, and 69.85 ng/μl (26.6–1,147 ng/μl; seven samples) for terrestrial mammals. We found that library preparation reduced the DNA concentration by approximately 53.81% for most samples; however, in some instances this value increased compared to the initial concentration of the extracted DNA (Table 1). In most sequencing runs we used Flongle flow cells with more than fifty pores available for immediate sequencing, nonetheless, in three Flongles, the pores were clogged, most likely because the use of cell priming and library loading reagents from the Ligation Sequencing Kit stored in plastic vials, as opposed to the recommended use of reagents stored in the glass vials of the Flongle expansion kit. This means that 21 sequencing runs were successful, while experiments EAFIT-M2126, EAFIT-M2153, and EAFIT-M2154 had insufficient or no sequencing output, and therefore, they were not considered in subsequent analyses. Our sequencing yield varied between 133.35 and 921.72 Mbp and was directly proportional to the number of available pores for immediate sequencing but not to the DNA library concentration (Fig. 3, Table 1). After quality filtering, an average of 82.58% of the reads from all our sequencing runs have a Q score ≥ 7, which represents between 30,019 and 357,353 reads. Finally, the recovered mtDNA fraction was low, ranging between 0.07% and 0.05%, for Minimap2 and mtBlaster analyses, respectively.

Figure 3 Sequencing output (Mbp) as a function of available pores and concentration of DNA extraction of the 21 individuals sequenced in this study.

Available pores are shown in black dots and the concentration of DNA extraction in blue diamonds.

Mitogenome reconstruction

We implemented a total of 252 assembly runs, resulting in 176 successful assemblies (complete and incomplete mitochondrial genomes), from which 100 yielded complete mitochondrial genomes (Table S2). Rebaler (reference-based assembler) recovered a higher number of assemblies regardless of the data type (WGS, Minimap2, mtBlaster) and was able to reconstruct 90.47% of the 21 possible mitogenomes, while Flye (de novo assembler) was only able to assemble 50% (Fig. 4, Table 2). On the other hand, Flye assembled the majority of mitogenomes of sequencing runs with more than 550 Mbp, but in sequencing runs with outputs below 550 Mbp Flye was only able to assemble 31.2% of the genomes, regardless of data type (Table 2). In contrast, Rebaler does not seem to be affected by the amount of sequencing yield as it assembled 89.6% of mitogenomes, even in runs with less than 550 Mbp regardless of data type (Table 2). Nonetheless, although Rebaler assembles a larger number of mitogenomes, we found that Flye performs better given the lower number of assembly errors (Fig. 4A, Table 3). On average, for all data types, Flye has 6.7% more assembly hits, while Rebaler on average has 15.4% more assembly errors (Table 3, S3). Our error composition comparison shows a higher number of errors in those assemblies obtained from mtBlaster-filtered data (Fig. 4B, Table S3). Overall, assemblies reconstructed with WGS and Minimap2-filtered data have a similar error composition; however, WGS genomes have a better quality than those obtained from a mtDNA filtering processes (mtBlaster or Minimap2) (Fig. 4A, Table S3). If the unassembled genomes are not considered, the mtBlaster+Rebaler pipeline generated the poorest quality assemblies, while those reconstructed using WGS+Flye had the best quality. Therefore, Flye outperformed Rebaler—in terms of assembly quality—in the genomes that were able to reconstruct (Figs. 4A, 4C). Noteworthy, we found that polishing with Nanopolish had a homogeneous effect for all assemblies with no quality improvement, regardless of data type or assembler used (de novo, or reference-based) (Fig. 4D, Table S4).

Figure 4 Quality evaluation of mitochondrial DNA reconstruction pipelines.

(A) Hit composition of the recovered assemblies, (B) error composition including the non-assembled category, (C) error composition without the non-assembled category, and (D) error composition including polished assemblies with Nanopolish (NP) versus non-polished treatments.

Table 2 Percentage of mitogenomes assembled by Flye and Rebaler for each data type, according to sequencing yield range.

Assembler	Sequencing yield (Mbp)	% Assembled mitogenomes	
		WGS	Minimap2	mtBlaster	
Flye	≥550	77.8	77.8	66.7	
	<550	25.0	33.3	33.3	
Rebaler	≥550	88.9	100.0	88.9	
	<550	91.7	83.3	91.7	

Table 3 Assembly quality evaluation based on the assembler and dataset treatment used.

Assembler	WGS	Minimap2	mtBlaster	
Assembly hits		
Flye	8.57% 27.75	5.47% 27.23	6.06% 26.05	
Rebaler	25.37	25.74	24.47	
Assembly errors	
Flye	9.25	9.77	10.95	
Rebaler	20.47% 11.63	13.23% 11.26	12.60% 12.53	
Note:

Assembly hits (coding genes and non-coding genes) and assembly errors (incomplete, null, and stop codon) excluding non-assembled genes of the mitogenomes assembled by Flye and Rebaler for each data type. The percentage corresponds to the increment of assembly hits (bold) or assembly errors (underline).

The mitochondrial genome depth (X) exhibited variations depending on the data type and assembler. In general, for WGS+Flye assemblies, the depth ranged from 3X to 17X (Table S2), while for genomes obtained through the WGS+Rebaler pipeline, the average depth ranged from 0.04X to 55.59X (Table S2). It is worth noting that pipelines based on mitochondrial reads filtering tended to yield higher average depth values than the WGS-based genome assemblies (Table S2). For example, the assemblies obtained through Minimap2+Flye showed depth values ranging from 4.16X to 63.17X, and those from Minimap2+Rebaler had average depths between 2.42X and 140.53X (Table S2). Likewise, genomes obtained from the mtBlaster+Flye pipeline ranged between 3.09X and 57.66X, and mtBlaster+Rebaler between 2.92X and 95.71X (Table S2).

Species identification based on phylogenetic analyses

From our assembled mitochondrial genomes, we were able to obtain 15 Cytb (12 species) and 13 COI (10 species) good-quality sequences susceptible to be included in phylogenetic analyses (Tables 4 and S1). For individuals of the genus Carollia (EAFIT-M2157 and EAFIT-M2168) only good-quality Cytb sequences were obtained, and the species (morphologically identified) Anoura geoffroyi, Glossophaga soricina, Molossus molossus, Sturnira bogotensis (one individual), and Zygodontomys brevicauda lacked sufficient quality for both mitochondrial markers (showed stop codons) and were excluded from the phylogenetic analyses (Table S1). We obtained nine Cytb datasets (alignments) which include 2,578 sequences (File S2) distributed in nine genera (ingroups). The sequence lengths range between 180 and 1,143 bp, resulting in 87.94% nucleotide coverage (Table S1). Similarly, for COI we obtained eight datasets, for eight different genera (ingroups), which include 2,395 sequences (File S2). COI sequences range in length between 201 and 657 bp, representing an average nucleotide coverage of 98.48% (Table S1).

Table 4 Results of our species identification pipeline at three taxonomic levels (Metazoa, order, family) and our morphology and phylogenetic systematics identifications.

		Metazoa-level	Order-level	Family-level				
Museum number	Number of mt reads	Number of hits	% reads correct Genus	% reads correct species	Metazoa ID	Percentage of Identity	Number of hits	% reads correct Genus	% reads correct species	Order ID	Percentage of Identity	Number of hits	% reads correct Genus	% reads correct species	Family ID	Percentage of Identity	Morphological ID	Phylogenetic ID CytB	Phylogenetic ID COI	
EAFIT-M 2126	141	17	64.71	47.06	Myotis riparius	93.29	12	91.67	66.67	Myotis riparius	93.29	12	100.00	66.67	Myotis riparius	93.29	Myotis riparius	Myotis riparius	Myotis riparius	
EAFIT-M 2106	187	44	65.91	0.00	Nephelomys albigularis*	96.39	41	70.73	0.00	Nephelomys albigularis*	96.39	41	70.73	0.00	Nephelomys albigularis*	96.39	Nephelomys pectoralis	Nephelomys pectoralis	Nephelomys pectoralis	
EAFIT-M 2143	92	20	65.00	5.00	Histiotus macrotus	91.80	13	100.00	7.69	Histiotus macrotus	91.81	13	100.00	7.69	Histiotus macrotus	91.81	Histiotus humboldti	Histiotus humboldti	Histiotus humboldti	
EAFIT-M 2107	127	21	47.62	47.62	Handleyomys alfaroi	96.20	19	52.63	52.63	Handleyomys alfaroi	96.20	19	52.63	52.63	Handleyomys alfaroi	96.20	Handleyomys alfaroi	Handleyomys alfaroi	Handleyomys alfaroi	
EAFIT-M 2124	365	32	53.13	15.63	Histiotus macrotus	96.36	20	85.00	25.00	Histiotus macrotus	96.36	21	80.95	23.81	Histiotus macrotus	96.36	Sturnira bogotensis	Defficient quality sequence	Defficient quality sequence	
EAFIT-M 2116	162	18	66.67	55.56	Carollia perspicillata**	96.53	13	92.31	76.92	Carollia perspicillata**	96.53	13	92.31	76.92	Carollia perspicillata**	96.526	Carollia brevicauda	Carollia brevicauda	Defficient quality sequence	
EAFIT-M 2160	547	45	35.56	11.11	Histiotus macrotus	93.49	32	56.25	15.63	Histiotus macrotus	93.49	34	55.88	17.65	Histiotus macrotus	93.49	Anoura geoffroyi	Defficient quality sequence	Defficient quality sequence	
EAFIT-M 2157	194	17	64.71	64.71	Artibeus lituratus	97.54	11	100.00	100.00	Artibeus lituratus	97.54	11	100.00	100.00	Artibeus lituratus	97.54	Artibeus lituratus	Artibeus lituratus	Artibeus lituratus	
EAFIT-M 2164	99	3	33.33	33.33	Glossophaga soricina	99.25	1	100.00	100.00	Glossophaga soricina	99.25	1	100.00	100.00	Glossophaga soricina	99.25	Glossophaga soricina	Defficient quality sequence	Defficient quality sequence	
EAFIT-M 2158	441	28	53.57	50.00	Artibeus lituratus	95.21	17	88.24	82.35	Artibeus lituratus	95.21	17	88.24	82.35	Artibeus lituratus	95.21	Artibeus lituratus	Artibeus lituratus	Artibeus lituratus	
EAFIT-M 2167	195	9	33.33	11.11	Sturnira bogotensis	95.66	3	100.00	33.33	Sturnira bogotensis	95.66	4	100.00	25.00	Sturnira bogotensis	95.66	Sturnira bogotensis	Sturnira bogotensis	Sturnira bogotensis	
EAFIT-M 2166	323	19	68.42	47.37	Carollia perspicillata**	97.60	13	100.00	69.23	Carollia perspicillata**	97.60	13	100.00	69.23	Carollia perspicillata**	97.60	Carollia brevicauda	Carollia brevicauda	Defficient quality sequence	
EAFIT-M 2165	817	101	21.78	4.95	Nephelomys childi	92.66	81	28.40	7.41	Nephelomys childi	92.66	86	27.91	6.98	Nephelomys childi	92.66	Nephelomys childi	Nephelomys childi	Nephelomys childi	
EAFIT-M 2168	287	24	25.00	25.00	Myotis nigricans**	93.32	8	100.00	75.00	Myotis nigricans**	93.32	8	100.00	75.00	Myotis nigricans**	93.32	Myotis caucensis	Myotis caucensis	Myotis caucencis	
EAFIT-M 2151	569	71	14.08	9.86	Nephelomys childi	93.48	62	32.26	22.58	Nephelomys childi	93.70	59	33.90	23.73	Nephelomys childi	93.70	Nephelomys childi	Nephelomys childi	Nephelomys childi	
EAFIT-M 2159	390	22	36.36	0.00	Histiotus macrotus	94.67	22	36.36	0.00	Histiotus macrotus	94.67	22	40.91	0.00	Histiotus macrotus	94.67	Molossus molossus	Defficient quality sequence	Defficient quality sequence	
EAFIT-M 2150	788	119	57.98	9.24	Akodon sp.**	97.09	103	63.11	10.68	Akodon sp.**	97.09	106	53.77	10.38	Akodon affinis	95.841	Akodon affinis	Akodon affinis	Akodon affinis	
EAFIT-M 2152	157	50	46.00	10.00	Marmosa robinsoni isthmica	93.01	49	28.57	14.29	Marmosa robinsoni isthmica	93.01	49	28.57	14.29	Marmosa robinsoni isthmica	93.01	Marmosa isthmica	Marmosa isthmica	Marmosa isthmica	
EAFIT-M 2161	464	136	30.88	2.21	Marmosa robinsoni	92.38	155	21.29	3.87	Marmosa robinsoni isthmica	92.21	155	21.29	3.87	Marmosa robinsoni isthmica	92.21	Marmosa isthmica	Marmosa isthmica	Marmosa isthmica	
EAFIT-M 2153	384	35	0.00	0.00	Histiotus macrotus	92.28	25	0.00	0.00	Histiotus macrotus	92.28	26	0.00	0.00	Histiotus macrotus	92.28	Zygodontomys brevicauda	Defficient quality sequence	Defficient quality sequence	
Notes:

For each dataset, the number of mitochondrial reads is shown. At every level of analysis, the number of total hits recovered using mtBlaster is included. Species with bolded letters (at the three levels of analysis) correspond to identifications that match the other two lines of evidence (i.e., morphology and phylogenetic analyses). Underlined text correspond to specimens for which the morphological and phylogenetic identifications (When available) did not match the one obtained through our species identification pipeline.

* Species lacking mitochondrial sequences in GenBank as of March 2023.

** Species where the most reliable hit corresponds to a misidentified GenBank sequence.

The Cytb and COI topologies are consistent with the literature and show in most cases monophyletic clades currently recognized as species or subspecies (Gutiérrez, Jansa & Voss, 2010; Moratelli et al., 2013; Velazco & Patterson, 2013; Almendra et al., 2018; Giménez, Giannini & Almeida, 2019; Rodríguez-Posada et al., 2021), but clades that exhibit paraphyletic “species” (see Myotis) are just the result of the current lack of resolution on their systematics (Larsen et al., 2012). Noteworthy, all of the Cytb and COI phylogenetic analyses match the morphological identification (Table 4). For our Cytb analyses, sequences of the species Akodon affinis, Artibeus lituratus, Carollia brevicauda, Histiotus humboldti, Myotis riparius, Sturnira bogotensis, Marmosa isthmica, and Handleyomys alfaroi, each formed a strongly supported monophyletic group with other sequences of the same species (Fig. 5). The species Myotis caucensis, Nephelomys childi, and N. pectoralis were not part of monophyletic groups with other conspecifics because there are no available Cytb sequences of these species in the Genbank and consequently they represent new genetic data for these lineages (Fig. 5). For the COI marker, the species Artibeus lituratus, Myotis caucensis, M. riparius, Sturnira bogotensis, Handleyomys alfaroi, Nephelomys childi, and N. pectoralis are part of haplogroups with other conspecific sequences, while Histiotus humboldti, Akodon affinis, and Marmosa isthmica represent the first available sequence for each species (Fig. 6).

Figure 5 Maximum Likelihood reconstructions of the Cytb mitochondrial marker for all the genera represented in our taxon sampling.

(A) Akodon, (B) Artibeus, (C) Sturnira, (D) Handleyomys, (E) Nephelomys, (F) Histiotus, (G) Carollia, (H) Marmosa, (I) Myotis. Terminals of non-collapsed clades include GenBank accession numbers. Phylogenies that exhibit paraphyletic species show the corresponding localities of each sequence. Colored terminals correspond to our sequenced material. Bootstrap support values above 75% are shown above branches.

Figure 6 Maximum Likelihood reconstructions of the COI mitochondrial marker for all the genera represented in our taxon sampling.

(A) Artibeus, (B) Handleyomys, (C) Marmosa, (D) Myotis, (E) Akodon, (F) Sturnira, (G) Histiotus, (H) Nephelomys. Terminals of non-collapsed clades include GenBank accession numbers. Phylogenies that exhibit paraphyletic species show the corresponding localities of each sequence. Colored terminals correspond to our sequenced material. Bootstrap support values above 75% are shown above branches.

Species identification pipeline with ONT raw reads

In 20 of the analyzed datasets, it was possible to recover mitochondrial reads, a single dataset was excluded from subsequent analyses (EAFIT-M2124 corresponding to the species Rhipidomys latimanus) because mtBlaster was unable to recover mitochondrial reads (Table S2). Our species identification pipeline did not show any relevant difference between the three taxonomic levels of analyses with respect to the “top” (most reliable) identification (Table 4). In all three analyses (Metazoa, order, family), for 10 samples, our pipeline matched the identification of the morphological+phylogenetic analyses (species Artibeus lituratus, Glossophaga soricina, Myotis riparius, Sturnira bogotensis, Handleyomys alfaroi, Nephelomys childi, Marmosa isthmica), four other samples were recovered by our pipeline with the correct genus but the species was incorrect due to misidentification of the sequence in the DNA library (species Carollia brevicauda, Myotis caucensis, Akodon affinis, Nephelomys pectoralis), and for one sample we also recovered the correct genus but the species was incorrect because of the lack of sequences of such species in the mtDNA library (Nephelomys pectoralis) (Table 4). Conversely, for five samples our pipeline recovered an incorrect identification even though the mtDNA library contains sequences of each species; however, from those four samples, only one recovered the incorrect genus and species (Handleyomys rhabdops instead of the correct Zygodontomys brevicauda), but the remaining four were allocated in the correct genus (Anoura, Histiotus, Molossus, Sturnira) (Table 4).

We also evaluated the effectiveness of the species identification pipeline for the genus and species by calculating the percentage of reads corresponding to the correct genus and species for each dataset at all levels of analyses (Metazoa, order, family). For our three levels of analyses, only three samples did not show any reads associated with the correct identification of the species; however, one of those datasets did not contain mitochondrial sequences for the species (Nephelomys pectoralis) whereas the other two datasets were unable to associate the reads with its correct species (Molossus molossus, Zygodontomys brevicauda) (Table 4, Fig. 7). For the other remaining 17 samples, all analyses—at all three levels—included reads associated with the correct species (Table 4). Overall, the analyses showed a consistent pattern, they were more effective at providing the correct identification as more taxonomic information was available, that is, a higher number of correct reads for the order and family databases than for the Metazoa database (Table 4, Fig. 7). We found that 10 of the order and family datasets provided more than 80% of the reads with the correct genus identification and seven datasets with at least 65% of their reads associated with the correct species (Fig. 7). There seems to be an additional pattern, the more sequences of the correct genus and species in the databases, the greater the number of reads associated with the correct species (Fig. 7). Generally, mitochondrial datasets (order and family) with more than 812 sequences provided >80.09% of correct identifications at the genus level, and datasets larger than 97 sequences produced >66.67% of correct identifications at the species level (Fig. 7).

Figure 7 Accuracy comparison of our species identification pipeline based exclusively on ONT reads, using three different taxon-level databases (Metazoa, order, family).

Percentage of reads that match the correct identification of the genus (left) and species (right) based on our morphological and phylogenetic systematics identification. The size of the dots indicate the number of available NCBI sequences for each genus (left) and species (right).

We observed variations in the processing time among the three taxonomic levels of analyses. When conducting a BLAST search without any prior sample identification (i.e., Metazoa), the analysis took between 01:26:37 and 01:38:57 h for datasets with the smallest and highest number of potential mitochondrial reads respectively (92 and 817 reads). In contrast, the analysis against the order-level dataset required only between 00:00:08 s and 00:01:28 min, while the family-level analysis took between 00:00:05 to 00:00:32 s. These findings highlight the substantial differences in computational time required depending on the database size and the relevance of additional taxonomic information when performing these identifications.

Discussion

Mitogenome reconstruction

In this study, we evaluated various computational routines for reconstructing mitochondrial genomes from non-model organisms, exclusively using ONT reads and found that this sequencing technology still exhibits limitations in comparison to other technologies. It maintains a higher raw sequencing error rate, approximately 5% (with the now deprecated R9 chemistry), which is notably higher than the approximately 0.3% error rate seen in Illumina (Baeza & García-De León, 2022). These errors are mainly attributed to mismatches between the raw signal and the basecalling process, generating modified nucleotides and homopolymer sequences (Magi, Giusti & Tattini, 2017). To address this issue, progress has been made in basecalling algorithms such as Guppy, improving the accuracy of basecalling compared to the older Albacore software (Baeza, 2020; Baeza & García-De León, 2022). Another strategy is based on error correction by implementing polishing programs specialized to deal with noisy long reads (Filipović et al., 2022). However, consistent with previous findings (Baeza & García-De León, 2022), we did not observe any noticeable improvement in the quality of the assemblies after applying a round of polishing with Nanopolish (Fig. 3).

The error rate limitations of ONT can also be effectively addressed through strategies such as maintaining a high read depth (Pollard et al., 2018; Ramón-Laca, Gallego & Nichols, 2023). Studies using PCR-free genome skimming approach with long reads, to reconstruct complete mitochondrial genomes have reported varying average genome depths, between 9X and 78X (Malukiewicz et al., 2021; De Vivo et al., 2022; Franco-Sierra & Díaz-Nieto, 2020). In our study, although most genomes assembled with either Flye or Rebaler exhibited depth values that fell within the range observed in the previously mentioned studies, only three routines produced high quality and error-free genomes: Nephelomys childi EAFIT-M2150 (17.00X), EAFIT-M2151 (36.62X), and Akodon affinis EAFIT-M2161 (26.09X), with NCBI accession numbers PP556414, PP555285 and PP556413 respectively (Table S2). We also observed that mitochondrial filtering increased depth values regardless of the assembler or filtering pipeline used (Table S2). This could be a signal that when the primary objective is to obtain good quality mitogenomes, filtering out the mitochondrial reads can increase the depth and thus the quality of the recovered genome (Ramón-Laca, Gallego & Nichols, 2023). Although depth is a key factor for reconstructing good quality genomes, our species identification pipeline does not heavily depend on this variable. Many datasets with shallow depths provided accurate identifications. For instance, the highest depth for the assembly of dataset EAFIT-M2158 was 4.72X (mtBlaster+Rebaler); however, this dataset produced correct identifications at all three levels of analyses (see below). However, although mtDNA filtering can increase genome depth, our analyses show that filtering with Minimap2 or mtBlaster does not necessarily increase the number of successful assemblies, in fact, we observe an increase in the number of errors such as incomplete genes, null genes, and premature stop codons (Fig. 4). This phenomenon may occur because the implemented algorithms fail to obtain all the mitochondrial reads or might recover erroneous mitochondrial sequences, such as NUMTs.

As for the assemblers, Flye is more accurate and has better quality overall; however, it requires higher sequencing throughput (≥550 Mpb) to assemble genomes, compared to Rebaler which, although it is able to assemble most mitogenomes, even from datasets with limited sequencing yield (<550 Mbp), it exhibits higher error rate (Table 2). Indeed, recent studies that have conducted comparisons of the resulting mitogenomes using these two assemblers have found similar results (Baeza, 2020; Baeza & García-De León, 2022). This phenomenon may have two different explanations. On the one hand, Rebaler uses Minimap2 for mapping potential mitochondrial reads to reference genomes, which has been shown to be difficult and leading to a decrease in the quality of resulting assemblies (Baeza, 2020; Baeza & García-De León, 2022). On the other hand, Flye implements a graph-based approach for single-molecule sequencing data that enhances genome assembly quality of noisy long-reads, like the ones generated by ONT (Kolmogorov et al., 2019; Zhang et al., 2022). Additionally, the option for metabarcoding assemblies included in Flye seems to work well in genomes that have low or uneven coverage (Kolmogorov et al., 2019), which is the case of mitochondrial genomes from whole genome shotgun sequencing.

Species identification pipeline with ONT raw reads

The implemented methodology exhibits promising results in the field of species identifications without the need of industrious laboratory work (no PCR), or expert knowledge on taxonomy and phylogenetic systematics; however, it will still need to optimize two main aspects before it can be extensively implemented for species identification. First, our methodology employs a Flongle flow cell (non-reusable; 100 USD) for sequencing, which yields up to 900 Mb of data and enables the identification of a single individual. However, the mtDNA fraction for species identification constitutes only 0.07% of the total sequencing output, resulting in most of the economic expenditure being allocated to sequencing data not used for identification purposes. Second, our methodological approach is not yet real-time, as it needs a minimum of 25 h to achieve species identification (24 h of sequencing plus <1 h for the species identification pipeline). Both problems can be addressed by using the standard and reusable Flow Cell (up to 50 Gb). This Flow Cell enables sequencing of the necessary output for individual species identification. It can be washed and reused for subsequent sequencing experiments, reducing sequencing costs per individual. In cases where time constraints are significant, it may be preferable to choose a Flow Cell that can produce the minimum sequencing output required for identification in a shorter time frame.

Our pipeline was able to correctly identify most (95%) of the samples to the genus level and half of the samples (50%) to the species level, but this last value can be up to 75% if we correct the identification of the sequences from BOLD or NCBI (Table 4). We found that species identification effectiveness was partially dependent on the sequencing output (Fig. 3). Overall, datasets that exceeded 550 Mbp, had more effective identifications compared to those that were below this sequencing yield. In some cases, not only the amount of data but the quality of the reads was related to the inability of our pipeline to accurately assess the taxon identity. In fact, in datasets containing low-quality reads, we encountered challenges not only in obtaining effective identifications, but also in extracting mitochondrial markers with sufficient quality for phylogenetic reconstructions as was the case for datasets EAFIT-M2160, EAFIT-M2164, EAFIT-M2152, and EAFIT-M2162 (Table 4).

Accuracy of identifications at the species and genus-level seems to be dependent on the availability of sequences in DNA repositories (GenBank, BOLD). Our results show that as more mitochondrial DNA data is available, there is a noticeable increase in the frequency of successful matches (blasted reads) accurately identifying the correct genus and species (Fig. 7). For instance, at the three levels of analysis (Metazoa, order, family) our species identification pipeline identified our N. pectoralis individual (dataset EAFIT-M2143) as N. albigularis, which can be explained by two facts. First, as of March 2023, there were no N. pectoralis sequences available in GenBank or BOLD. Second, N. pectoralis is an endemic species from Colombia, recently split from N. albigularis which is currently recognized as a species restricted to Peru and Ecuador (Weksler, Percequillo & Voss, 2006), thus, it makes sense that the most reliable identification is from a closely related lineage.

Meanwhile, automated species identification methods, including our approach, depend heavily on the identification—scientific name—assigned to sequences from repositories like BOLD and GenBank. Consequently, misidentifications or outdated names in the sequences from DNA databases will obviously produce incorrect identifications. Three cases in our analyses illustrate this scenario. Our two individuals morphologically identified as Carollia brevicauda, were defined by our species identification pipeline as C. perspicillata. Historically, these two species have been frequently misidentified due to their morphological similarities (Ruelas & Pacheco, 2022). However, we were able to obtain high-quality mitochondrial markers for both datasets and their analyses with other congeneric sequences confirmed our initial morphological identification. Moreover, while the “top” identification retrieved through the species identification pipeline was C. perspicillata, most of the recovered hits corresponded to C. brevicauda (File S1). Therefore, although we were unable to examine the vouchers, we are inclined to believe that the specimen from GenBank corresponding to the most reliable hit is an example of misidentification.

A similar example occurs with the individual EAFIT-M2159, morphologically identified as Myotis caucensis, which was identified by our species identification pipeline as M. nigricans. Our morphological identification was based on the original description of the species (Moratelli et al., 2013). We found that although dorsal and ventral coloration patterns follow more closely the description of M. handleyi, dental characters are more closely related to M. caucensis (Moratelli et al., 2013). In particular, the upper second premolar which is evidently visible in lateral view on the latter species, and is also present in our material (Moratelli et al., 2013). Additionally, M. handleyi is a recently described species believed to be sister to M. caucensis (Moratelli et al., 2017), consequently, molecular data uploaded to databases before these taxonomic updates may not accurately reflect the current taxonomy. Sequences previously labeled as M. nigricans could pertain to either M. handleyi or M. caucensis. In fact, our phylogenetic analysis of Cytb supports this hypothesis, as our sequence is sister to M. handleyi (Fig. 5). In conclusion, the absence of updated taxonomy associated with sequences poses challenges for the use of genetic data within automated species identification pipelines, such as the one presented here.

The third example corresponds to an individual of the species Akodon affinis (EAFIT-M2161). Using the Metazoa and order databases, our sample could not be identified to the species level because the most reliable hit in both cases corresponds to the name Akodon sp. (NCBI:txid2964413). Two lines of evidence lead us to believe that this last specimen corresponds to the species A. affinis. First, the specimen was collected in Cauca (Colombia), a locality reported to be within the distribution range of A. affinis (Patton, Pardiñas & D’Elía, 2015). Second, our phylogenetic analysis supported our morphological identification, as our sequence formed a highly supported monophyletic group with another A. affinis sequence (Fig. 5). As in the two previous examples, misidentification decreases the effectiveness of our species identification pipeline.

The mtDNA “in silico enrichment” strategy which uses DC-BLAST to isolate mitochondrial reads from WGS sequencing data, proposed by Franco-Sierra & Díaz-Nieto (2020) and optimized here, was highly efficient in reducing computational time and increasing the effectiveness (accuracy) of species-level identifications. Consequently, the availability of multiple mitochondrial markers for diverse metazoan organisms allows to simplify species identification process by only using the mitochondrial fraction of the sequenced genome (Hebert, Ratnasingham & de Waard, 2003; Stoekle, 2003; Page & Hughes, 2010; Armani et al., 2017; Elyasigorji et al., 2023). Noteworthy, our results indicate that the computational pipeline for species identification can be more efficient, but only slightly more accurate, depending on the size of the database implemented for DC-BLAST (Metazoa, order, family). This is because a smaller reference database reduces the amount of computational load, but the gain in accuracy was only reduced to one (see Akodon affinis) out of 20 analyses.

Furthermore, automated species identification methods such as the one used in this study should always undergo a critical review process to analyze their results. Although an e-value very close to zero indicates a high level of confidence in a particular identification (Kerfeld & Scott, 2011; Shah et al., 2019), it is important to consider other output parameters. For example, it is important to account for the read length, as short reads can produce very low e-values because the shorter the sequence the more likely it is to match the nucleotides of a reference sequence (Kerfeld & Scott, 2011; Shah et al., 2019). Although short reads (ca. 100 bp) can be useful for identification purposes in a phylogenetic scenario (Díaz-Nieto, Jansa & Voss, 2016; Teta & Díaz-Nieto, 2019), their usefulness seems to be reduced with alignment search algorithms. It is therefore important to examine the proposed identification considering basic natural history information such as distribution and habitat, to determine the validity and meaningfulness of the recovered identification, as explained with the Nephelomys pectoralis case.

It is clear that our species identification pipeline is not infallible, and it will likely be benefited from the new ONT chemistry (Ligation Sequencing Kits SQK-LSK114 and Flongle Flowcells R10) as it significantly increases the quality of the sequenced reads from ~90% to ~99% (Sanderson et al., 2023; Ni et al., 2023). However, to further improve the effectiveness of our pipeline it will be important to increase the recovered mitochondrial fraction, which although concordant with other studies (Franco-Sierra & Díaz-Nieto, 2020; De Vivo et al., 2022) it is still very small (between 0.01% and 0.06% of total bases). Multiple methodologies have been developed to increase the mtDNA fraction in a sample, such as the use of centrifugation protocols (Stockburger et al., 2016; Gaudioso et al., 2019; Ramón-Laca, Gallego & Nichols, 2023), PCR-based methodologies (Zascavage et al., 2019) and Cas9 approaches (Ramón-Laca, Gallego & Nichols, 2023). Nonetheless, additional laboratory steps include specialized equipment (ultracentrifuge) or enzymes whose life depends on constant refrigeration, hinder the implementation of this sequencing technology under field conditions, which we consider is the ultimate goal with this technology. Also, incorporating more reagents and equipment would increase operating costs, reducing its accessibility for institutions with limited budgets. Although, the overall cost of mitochondrial genome sequencing decreases with the implementation of mitochondrial DNA enrichment or sample multiplexing (Zascavage et al., 2019; Ramón-Laca, Gallego & Nichols, 2023), such approaches require expensive materials and reagents that will intrinsically inflate the protocol costs. Finally, increasing the mitochondrial fraction may provide more accurate identifications, but also augment sample processing time in up to 48 h depending on the protocol (Zascavage et al., 2019; Ramón-Laca, Gallego & Nichols, 2023), reducing the “real-time” advantage of this approach.

Species identification based on phylogenetic analyses

Our results make an important contribution to the knowledge of small mammal diversity, but also help to fill the gaps that still exist in genomic databases. This project is generating the first complete mitochondrial genomes for the species Akodon affinis and Nephelomys childi. Additionally, the first COI sequences for five species (Akodon affinis, Sturnira bogotensis, Nephelomys pectoralis, Histiotus humboldti and Myotis caucensis); first Cytb sequences for the species Nephelomys childi, N. pectoralis, and Myotis caucensis; the first two (2) sequences for Nephelomys childi representing populations from Cordillera Central (Colombia), and the second Cytb sequence for the species Akodon affinis.

Our Cytb sequences for Marmosa isthmica form a highly supported group sister to the known M. isthmica haplogroup with an uncorrected p-distance of 0.068 between clades (Fig. 5). Interspecific distances between congeneric species are similar and even smaller, as is the case of M. demerarae and M. merida or M. parda and M. rutteri with p-distance values of 0.07 and 0.05, respectively (Voss et al., 2020). Marmosa isthmica is distributed from western Panama along the Pacific coast of Colombia to Santa Rosa (Ecuador), including the lowlands of northwestern Colombia, and the intern-Andean valleys of Cauca and Magdalena rivers. Despite extensive revisions (Rossi, Voss & Lunde, 2010; Gutiérrez, Jansa & Voss, 2010; Gutiérrez et al., 2014; Voss et al., 2020), most studies have not included genetic information from Colombian localities (Gutiérrez, Jansa & Voss, 2010; Gutiérrez et al., 2014; Giarla & Voss, 2020). Noteworthy, our two M. isthmica individuals were collected in dry forest areas of the Cauca River, below 700 masl, whereas most collection sites reported for this species correspond to moist lowland and premontane forests below 1,700 m elevation (Rossi, Voss & Lunde, 2010). Consequently, it is important to evaluate whether the observed genetic distance between these two clades suggests the presence of cryptic diversity or is merely the result of an isolation-by-distance effect that will be diluted when an expanded geographical sampling becomes available.

Nephelomys childi is endemic to cloud forest habitats in the three Andean Cordilleras of Colombia (Patton, Pardiñas & D’Elía, 2015) and inter-Andean valleys have been proposed as barriers that prevent migration between populations resulting in the observed pattern of allopatric distributions in the three Cordilleras (Cárdenas González, 2017). We found that our COI sequences of N. childi form a highly supported clade sister to a haplogroup from the Cordillera Oriental (Fig. 7). We also observed a considerable variation (5.61% in uncorrected p-distance) between these two haplogroups (Fig. 7). Considering these facts, we may be facing two different scenarios. First, there is hidden diversity within the N. childi group, and the allopatric distributions reported for the species may have triggered allopatric speciation processes. In fact, this is further substantiated by recent systematic studies on this genus (Weksler, Percequillo & Voss, 2006), which have emphasized the need for a taxonomic revision to elucidate the internal phylogenetic relationships and possible presence of cryptic species, establish diagnosis, and clarify the geographical distribution of the species (Cárdenas González, 2017; Ruelas et al., 2021). In a second scenario, our sequence divergence is just the result of the geographic distance found between populations (Wright, 1943; Meirmans, 2012). The species N. childi is underrepresented in genomic databases. To date, the genus has 169 mitochondrial sequences available on GenBank, of which only five belong to N. childi (Cordillera Oriental) and are restricted to the marker COI. Incrementing the molecular sampling along the entire distribution of the species would close the gap in the genetic distance between populations of the species.

Conclusions

In this study, we explored the use of the ONT MinION device with the Flongle flow cells as a tool for the reconstruction of mitochondrial genomes of small mammals, and to obtain species identifications of non-model mammals from multiple lineages (opossums, rodents, bats) using solely ONT reads. Specifically, we learned that for sequencing runs above 550 Mbp, the mitochondrial genome reconstruction should be performed using pipelines based on de novo assemblies, which have a better quality regardless of data type. However, if sequencing runs have less than 550 Mb a better result can be obtained by using a referenced-based assembler. Furthermore, although mitochondrial DNA filtering from WGS data seems to increase the depth of recovered mitogenomes, it does not appear to increase the number of successful assemblies. Instead, mitochondrial DNA filtering produces more assembly errors compared to those obtained from WGS data. Also, our findings suggest that the polishing step is not required as it does not improve the quality of assemblies, instead it is time- and computational-resource consuming.

Although our pipeline has proven not to be flawless, we observed that it produces identifications with more than 95% accuracy in non-model species of Neotropical small mammals. This can become a powerful approach for the description and characterization of the planet’s diversity because it can be implemented by any researcher (even without extensive taxonomic, phylogenetic, or morphological training), in real-time and even under field conditions. Species identification results are highly dependent on the availability of data in public repositories and the curatorial (including taxonomy updates) of that information. We found that the more sequences of a taxonomic group are available in a database, the more effective is the identification of an individual with unknown identity. This demonstrates the importance of contributing to the growing field of species identification and description and closing the gap of knowledge on non-model organism diversity. Accordingly, the results obtained in this study make an important contribution to increasing the knowledge of the genetic diversity of small mammals by making available three new mitochondrial genomes and eight new DNA markers for science.

The findings of this work add to the growing body of data demonstrating the usefulness of ONT as a PCR-free option for sequencing full organellar genomes and obtaining species identifications in a short amount of time and on a wider taxonomic scale. Furthermore, the proposed methodology in this work stands as a relevant tool to help increase the available data for non-model organisms, and possibly it can be applied for onsite non-model species identification.

Supplemental Information

Supplemental Information 1 A detailed description of each sampling locality.

Supplemental Information 2 DC-Blast detailed results for each individual at each level of analysis (Metazoa, Order, Family).

Supplemental Information 3 Accession number and length of all sequences used to perform the phylogenetic analyses presented in this article.

Supplemental Information 4 Supplementary Tables.

Supplemental Information 5 CytB sequences.

Supplemental Information 6 COI sequences.

Supplemental Information 7 Complete mitochondrial genome EAFIT-M2161.

Supplemental Information 8 Complete mitochondrial genome EAFIT-M2151.

Supplemental Information 9 Complete mitochondrial genome EAFIT-M2150.

We acknowledge the Apolo Scientific Computational Center from EAFIT University and especially Laura Sánchez Córdoba for her constant support and providing access to generous computational resources. We thank Hacienda La Sierra, Universidad EAFIT, and Finca Socolao for their generosity facilitating our logistics during the field work. We thank Esteban Velásquez for his bioinformatic support during the initial stages of the development of this project, and Oscar S. Alzate, Juan M. Lozano, Luis J. Pérez, and Mariana Cruz from the Biodiversity research group for their help with fieldwork activities. We would also like to thank Y. Xilena Rueda for her assistance at the biological collection of Universidad EAFIT.

Additional Information and Declarations

Competing Interests

Author Contributions

Animal Ethics

Field Study Permissions

DNA Deposition

Data Availability

Nicolás D. Franco-Sierra is employed by Syndesis Health and is also an ad honorem member of VEDAS Corporación de Investigación e Innovación (VEDAS CII), a non-profit research organization.

Sara Velasquez-Restrepo conceived and designed the experiments, performed the experiments, analyzed the data, prepared figures and/or tables, authored or reviewed drafts of the article, and approved the final draft.

Mariana Corrales Orozco conceived and designed the experiments, performed the experiments, analyzed the data, prepared figures and/or tables, and approved the final draft.

Nicolás D. Franco-Sierra conceived and designed the experiments, authored or reviewed drafts of the article, and approved the final draft.

Juan M. Martínez-Cerón analyzed the data, prepared figures and/or tables, and approved the final draft.

Juan F. Díaz-Nieto conceived and designed the experiments, analyzed the data, authored or reviewed drafts of the article, and approved the final draft.

The following information was supplied relating to ethical approvals (i.e., approving body and any reference numbers):

The Ethichs committee of The Ethics committee of the Universidad EAFIT provided full approval for this research in the November 16th 2021 virtual session. Attached as Supplemental File is the ethics committee approval letter.

The following information was supplied relating to field study approvals (i.e., approving body and any reference numbers):

Field experiments were approved by the “Autoridad Nacional de Licencias Ambientales (ANLA) under resolution 1160 of June 7, 2023”.

The following information was supplied regarding the deposition of DNA sequences:

The COI and CytB sequences are available at GenBank: OR873409-OR873422; OR988047-OR988062.

The complete genomes are available at GenBank: PP556413, PP556414 and PP555285.

The genomes are also available in the Supplemental Files.

The following information was supplied regarding data availability:

The raw fastq sequences are available at FigShare: Velasquez, Sara; Díaz-Nieto, Juan F. (2024). Quality filtered data. figshare. Dataset. https://doi.org/10.6084/m9.figshare.24669249.v1.

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
