# Peer review of "Identification of non-model mammal species using the MinION DNA sequencer from Oxford Nanopore"

_PeerJ, doi:10.7717/peerj.17887_

## Round 0.1 · original submission · Major Revisions

· Academic Editor

Major Revisions

Both reviewers agree on some crucial points :

Citing previous studies on the subject, and the data needs to be made available, both mitogenomes in NCBI and raw data. At least the ones that are high quality need to be published in a sequence database, and the lower quality ones could be included with the raw data or as additional data. The reporting on mitogenome assembly quality needs to be detailed (not only averages, but also min-max values of coverage and quality for each), as it is central to the argument made in the paper.

More generally, there are a number of points to address. Reviewer 2 in particular has made a very detailed review with lots of excellent suggestions, including on the orientation of the introduction and paper itself. Indeed, the results, while valuable, do not support the current framing of the article for cheap and efficient barcoding in the field, as they use one flongle per sample. There have been a number of publications recently aiming to optimize molecular identification and data generation using flongles, and they manage to sequence hundreds of shorter ID sequences (barcodes mostly) per flongle, with much lower bioinformatics needs and deeper coverage (so higher sequence quality). Including large numbers of specimens would needed for answering taxonomic, distribution, and population dynamic questions, even in the examples listed by the authors. Even for the field aspect, the suggested protocol is still long, and might not be the best use of (generally too limited) field time, but this can be discussed.

Therefore, the paper needs to be reoriented to take into account the real results of the study; a criticism of the approach in general would be highly valuable too, as suggested by reviewer 2.

·

Basic reporting

.

Experimental design

.

Validity of the findings

.

Additional comments

The study by Sara Velásquez-Restrepo and collaborators is interesting and I think deserves to be published in PeerJ
It reads well, it is clear, proper writen expression, the methods are well explained and detailed.
Below, please, find my comments:


Introduction
Second Paragraph. I agree with the authors. However, what if you have sequencing facilities in your university or in a different university in your country. That is a reality in Colombia. You are describing an extremme situation. Please, town down the claims. Also, refer to time consumption. That is a main issue that you are not mentioning.

Line 138. Franco-Sierra & Díaz Nieto (2020) and De Vivo et al. (2022) are not the only studies doing that. Please, include additional authors doing the same.


Methods.

Line 183. I agree with the statement and sentence. But, you are not the first comparing long read to short read assembled mitogenomes. Please, give credit to previous studies doing that.


Results
Section 'Mitogenome reconstruction'
The authors are providing percentages here. It is not clear if the percentage refers to the number of species or mitogenome completeness per species. Please, improve this part. If number of species, provide numbers to make the information clear to the reader


Also, I suggest creating a new subsection to highlight how accurate are the assembled mitogenomes. That information is relevant but it is difficult to know how accurate they are really based on the information provided by the authors. Other than creating a new section, I suggest expanding this section of the introduction


Discussion
Interesting discussion. I found it a little too long. I suggest reducing it by 10% or so.

Conclusion
Again interesting, but too long.


I went to GenBank and attempted to check the mitochondrial genomes and raw datasets. They are not available there. Please, make sure to make them available immediately.


Good work! J. Antonio Baeza

Reviewer 2 ·

Excellent Review

This review has been rated excellent by staff (in the top 15% of reviews)
EDITOR COMMENT
This review is extremely complete and constructive. It identified strengths of the submission, suggested corrections for the weaknesses both in the results and in regard to the literature context, and also detailed smaller corrections. Following it should lead to considerable improvement of the paper. Thank you very much !

Basic reporting

This is a well-written and well-structured manuscript that highlights the use of ONT long-read sequencing for species identification.

While the manuscript is generally logical, I think the introduction is the weakest part of the manuscript because the authors make the case that ONT technology will speed up species identifications in the Neotropics where there is high species richness—that the work can be done in the field, without sacrificing the animal, without expensive equipment, and without species-specific primers. The flaws are that the authors did not perform that work in the field (it was done in a lab), they did end up sacrificing the animals, and there is a need for expensive equipment on the bioinformatics side of the workflow. The manuscript would be improved if the focus was on the use of long-reads datasets for species barcoding. While the results of this manuscript demonstrate that it is currently not an efficient way to barcode animals, I do think that this is an important result—for researchers to understand the current limitations. Thus, if the introduction was shifted to the state of ONT sequences for mitochondrial reads and the need for developing technology that is cost-effective, reliable, and can be performed in the field. I also see a lot of papers coming out on nanopore adaptive sampling and it may be worth mentioning in the introduction or discussion.

The figure legends could use more description, in general. For example, Figure 1 has five numbers that correspond to sampling localities; however, they are not mentioned in the description or the text of the paper. Figure 2 (Computational workflow) looks good. Figure 3 has a misspelling of Mbp in the description, but otherwise looks good. Figure 4, the font is a bit small, but otherwise looks good. Figure 5 looks good. In Figures 6 and 7, the scientific names need to be italicized. Also, in the text (line 175) there is a reference to Appendix 1, but I am not sure what that is referring to.

One thing to consider is the Collector Number “JFD number” is used on the raw reads and the manuscript figures, and most of the tables. However, Table 4 does not include the JDF number, nor does Table S2. I think it is easier to stick with one “id number” throughout.

Author disclosure indicates that the raw data will be publicly available via Figshare; however, the authors did not mention submitting the mitogenomes nor the single mitochondrial gene data to GenBank, even though one of the major points they make throughout the paper is the limitations of data available on NCBI for BLAST searches, I find this to be a weakness of the manuscript. Minimally, the data availability should be mentioned in the manuscript and authors should consider submitting to NCBI to support their argument for improving BLAST searches.

Experimental design

I think the strength of the paper lies in the experimental design, methods, and reporting of the bioinformatics workflow (e.g., the results of the pipeline reported in Figure 2). The methods are well-constructed and repeatable. I think this could be further improved if the authors clearly stated how their pipeline fills an important gap in the literature.

Validity of the findings

The experiments were well-designed and the interpretation of the results was sound. The only issues that appear in the conclusions are related to the issues I found in the introduction. For example, Lines 536-542 discuss “effective species-level identification without the need for industrious laboratory work that includes PCR, the need of expert taxonomists, or extensive knowledge on phylogenetic systematics”. This argument is flawed because I am not convinced that this is the most effective or cheapest way to barcode. Further, the laboratory needs will now shift to the need for computer servers and bioinformatics expertise. Also, and perhaps the weakest part of the argument is that you no longer need taxonomists or knowledge of phylogenetics—throughout the manuscript, the authors comment about how the lack of expert identifications has hampered their workflow and yet they refute this argument throughout the manuscript. For example”
Line 404: “lack of resolution on their systematics”
Lines 427-428: “ …but the species was incorrect due to misidentification of the sequence in the DNA library”
Line 567: “DNA databases will obviously produce incorrect identifications.”
Lines 590-592: “…the absence of updated taxonomy associated to sequences poses challenges…”
Lines 667-670: “…without the need of detailed knowledge of a taxonomic group…”
Lines 722-725: “…without the need for an expert taxonomist”
Lines 744-745: “Species identification results are highly dependent on the availability of data…including taxonomic updates.
The manuscript could be improved by presenting the results as “the current state of ONT long read sequences”. Readers will be interested to know the limitations of using this technology for species identification. At the moment, it is written like the Flongle is the solution to all of our taxonomy problems under field conditions when your text (see about) presents the contrary.

Additional comments

Minor suggestions:
Line 23: "challenging identification" change to "are difficult to identify using morphology"
Line 57: change "globe" to World or Earth
Line 82: remove the word "large
Line 85: spell out ID and change "exportation" to "transfer" or "transport"
Lines 96-99: I was unsure what this was referring to. Are you making the case to use multiple mitochondrial markers or nuclear ones?
Line 151 and Lines 186-187: the euthanization of animals argument is not going to work in this paper because you euthanized animals.
In the introduction I would also be careful with the export permit logic--I think ONT even has some rules for how their equipment is used under field conditions.
Line 175: Appendix 1?
Line 341: Mpb?
There seems to be a lot of formating errors in the literature cited section

---

## Round 0.2 · accepted · Accept

· Academic Editor

Accept

Thank you very much for addressing all the reviewers' comments in your revision. We have received a review of your revision by a previous reviewer and they are satisfied with the changes you have made.

Your manuscript is ready for publication now.

Reviewer 2 ·

Basic reporting

no comment

Experimental design

no comment

Validity of the findings

no comment

Additional comments

I read through the original manuscript as well as the revised version and I feel that the authors did an adequate job addressing my original concerns (reviewer 2). I am satisfied that they have met the requirements for publication.